# Cellular Senescence, Mitochondrial Dysfunction, and Their Link to Cardiovascular Disease

**DOI:** 10.3390/cells13040353

**Published:** 2024-02-17

**Authors:** Maria Camacho-Encina, Laura K. Booth, Rachael E. Redgrave, Omowumi Folaranmi, Ioakim Spyridopoulos, Gavin D. Richardson

**Affiliations:** 1Vascular Medicine and Biology Theme, Bioscience Institute, Newcastle University, Newcastle upon Tyne NE1 3BZ, UK; rachael.redgrave@newcastle.ac.uk (R.E.R.); o.folaranmi2@newcastle.ac.uk (O.F.); gavin.richardson@ncl.ac.uk (G.D.R.); 2Vascular Medicine and Biology Theme, Translational and Clinical Research Institute, Newcastle University, Newcastle upon Tyne NE1 3BZ, UK; l.booth2@newcastle.ac.uk (L.K.B.); ioakim.spyridopoulos@ncl.ac.uk (I.S.)

**Keywords:** senescence, mitochondrial dysfunction, cardiac cardiomyocyte, cardiac ischemia reperfusion

## Abstract

Cardiovascular diseases (CVDs), a group of disorders affecting the heart or blood vessels, are the primary cause of death worldwide, with an immense impact on patient quality of life and disability. According to the World Health Organization, CVD takes an estimated 17.9 million lives each year, where more than four out of five CVD deaths are due to heart attacks and strokes. In the decades to come, an increased prevalence of age-related CVD, such as atherosclerosis, coronary artery stenosis, myocardial infarction (MI), valvular heart disease, and heart failure (HF) will contribute to an even greater health and economic burden as the global average life expectancy increases and consequently the world’s population continues to age. Considering this, it is important to focus our research efforts on understanding the fundamental mechanisms underlying CVD. In this review, we focus on cellular senescence and mitochondrial dysfunction, which have long been established to contribute to CVD. We also assess the recent advances in targeting mitochondrial dysfunction including energy starvation and oxidative stress, mitochondria dynamics imbalance, cell apoptosis, mitophagy, and senescence with a focus on therapies that influence both and therefore perhaps represent strategies with the most clinical potential, range, and utility.

## 1. Cellular Senescence

Cellular senescence was originally defined as the irreversible exit from the cell cycle [1]. However, over recent years the definition of senescence has evolved and now includes many other characteristics such, as mitochondrial dysfunction, resistance to apoptosis, and the activation of a hypersecretory phenotype termed the senescence-associated secretory phenotype (SASP) [2]. Arguably, these characteristics more accurately define senescence given the substantive evidence that non-proliferative, post-mitotic cells, such as cardiomyocytes, can become senescent [3,4,5,6]. Similarly, while originally described as a consequence of telomere attrition following extensive proliferation [7,8], it is now accepted that numerous types of stresses which result in DNA damage within the genome or within the telomeres can activate pathways controlling senescence [9] (Figure 1). However, in both proliferative and post-mitotic cell populations, senescence induction is associated with the activation of either, or both, p21 and p16, cyclin-dependent kinase inhibitors that are components of the tumour suppressor pathways governed by the transcription factor p53 and the retinoblastoma protein (RB), respectively [10]. The traditional view that cellular senescence evolved as a tumour-suppressive mechanism has recently been challenged, as evidence suggests that senescent cells contribute to several important physiological processes throughout life, including tissue development, wound healing, and tissue repair [11]. In addition to these beneficial roles, cellular senescence has also been shown to be crucial in tissue pathophysiology, representing a key driver of ageing and age-related diseases [12,13,14]. Therefore, cellular senescence can be viewed as an example of antagonistic pleiotropy, which is a cellular program which is beneficial in one setting but deleterious in another [15]. 

Within the cardiovascular system, models of induced and attenuated senescence have implicated senescence in the pathophysiology of myocardial remodelling (age-related, chemotherapy-induced [16], and post-injury) [3,4,17,18,19], hypertension, atherosclerosis [20], and the development of aortic aneurysms [21], and have been extensively reviewed [22,23,24,25,26,27].

## 2. Mitochondrial Abnormalities in Cardiovascular Diseases

Mitochondria are double-membraned organelles with their own circular genome, mitochondrial DNA (mtDNA), which is replicated independently of the host genome [28]. Mitochondria are involved in diverse yet interconnected functions, including the production of adenosine triphosphate (ATP), and the regulation of nutritional metabolism, calcium homeostasis, and programmed cell death [29,30,31]. They are found in the cytoplasm of nearly all eukaryotic cells as highly dynamic networks, undergoing coordinated cycles of biogenesis, fusion, fission, and degradation (mitophagy) to sustain their homeostasis and to adapt energy production based on the cell’s needs [32]. Proper mitochondrial function and dynamics are particularly necessary in tissues and cells with high energy demands such as the heart, and particularly in cardiomyocytes, which continuously require ATP to sustain cardiac activity. In adult cardiomyocytes, mitochondria occupy nearly one-third of the total intracellular volume [33] and provide approximately 95% of the ATP consumed by the heart [34]. It is, therefore, unsurprising that functional abnormalities in cardiac mitochondria have emerged as a key factor in cardiovascular disease (CVD) leading to decreased ATP production and energy supply, increased reactive oxygen species (ROS) production, cell apoptosis, and mitochondrial dynamic imbalance [35].

### 2.1. Energy Starvation and Oxidative Stress

Decreased energy supply is considered to be a leading consequence of mitochondrial dysfunction. The heart’s voracious requirement for energy, in the form of ATP, mainly relies on oxidative phosphorylation (OXPHOS) or β-oxidation of fatty acids and the tricarboxylic acid (TCA) cycle in the mitochondria. During pathological myocardial remodelling, there is a reduction in the levels of carnitine in the heart [36], an essential cofactor for mediating the entry of fatty acids into the mitochondria at the site of β-oxidation [37]. Due to this reduction in fatty acids’ availability within the mitochondria, cardiac metabolism is reprogrammed towards increased reliance on glucose as the energy resource with a significant increase in glycolysis, to maintain ATP production. However, ATP generated from glycolysis contributes less than 5% of the total ATP consumed [36], which is not enough to compensate for the reduction in fatty acid oxidation, and therefore, cardiac ATP is progressively depleted. The role of energy deprivation in the induction and pathogenesis of heart failure (HF) is well supported by clinical evidence, in which therapeutic measures to reduce energy consumption have been demonstrated to improve survival while treatment increasing energy demand is detrimental [38]. In the mitochondria, the synthesis of ATP takes place in the electron transport chain (ETC) [39]. Reduced nicotinamide adenine dinucleotide (NADH) and reduced flavin adenine dinucleotide (FADH_2_) generated from the Krebs cycle, and from β-oxidation, transfer protons and electrons through the ETC, creating an electrochemical gradient that is then used to activate ATP synthase and produces ATP. Alterations in mtDNA genes such as NADH-dehydrogenase genes (*MT-ND1*, *MT-ND5* and *MT-ND6*), cytochrome b (*MT-CYB*), cytochrome c oxidase I and II (*MT-CO1* and *MT-CO2*), and ATP synthase 6 (*MT-ATP6*), have been described in dilated cardiomyopathies [40]. Reduced activities of complexes I and IV, as well as of the NADH phosphate (NADPH)-transhydrogenase and the Krebs cycle enzymes have been also observed in patients with HF [41,42]. Interestingly, some studies suggest that mtDNA mutations induce cardiovascular senescence and CVD, as demonstrated by the observation that *Polg^m/m^* mice, which are prone to the accumulation of mitochondrial DNA mutations, have increased expression of senescent markers *p16^ink4a^* and display early onset cardiomyopathy [43,44]. It remains unclear what mechanism mediates this induction, but an increase in mitochondrial ROS (mtROS) has been proposed as a causal factor. 

Mitochondrial ATP production is accompanied by the generation of ROS (Figure 2), a generic term for an array of short-lived and unstable free radicals that contain oxygen with vastly different properties and biological functions that range from signalling (when strictly regulated) to causing cell damage [45]. Physiologically, ROS-mediated signalling pathways are associated with cell survival and proliferation, combatting infectious agents, and have mitogenic effects on cells [46,47]. However, excessive ROS production drives oxidative stress, a deleterious process that potentially causes irreversible damage to various molecules and structures within the cell [48], leading to further mitochondrial dysfunction, oxidative stress, and cell death [30,49]. Increased ROS appears to be capable of inducing senescence through several mechanisms. Telomeres are particularly sensitive to ROS-induced damage, possibly due to their guanine-rich regions, which increase their susceptibility to oxidation [50,51], and increased ROS can accelerate telomere attrition contributing to telomere dysfunction, premature senescence, and accelerated ageing [52]. ROS also generate DNA lesions in the form of single-stranded DNA and/or double-stranded breaks (DSBs) within the genomic or telomeric DNA. Eventually, as a result of telomere shortening or DNA damage, activation of the DNA damage response (DDR) occurs [53]. The DDR is an evolutionarily conserved signal transduction pathway required for genome integrity preservation. It coordinates cellular efforts to repair DNA damage, which, if unsuccessful, directs cell fate towards apoptosis or senescence thereby impeding the propagation of corrupted genetic information [54]. The DDR is characterised by the recruitment and activation of two large protein sensor kinases at the site of the lesion: ataxia telangiectasia and Rad3-related (ATR) protein when single-stranded DNA is exposed, and ataxia-telangiectasia mutated (ATM) protein at DSBs. The recruitment of ATR or ATM to the lesion causes the local formation of DNA damage foci containing the phosphorylated form of the histone H2AX (γH2AX) and ultimately inducing cell-cycle arrest through the activation of checkpoint proteins, including p53 [55]. Furthermore, once senescent, cells exhibit a decreased mitochondrial membrane potential, increased proton leak, and enhanced production of mtROS [56]. As such, the elevated ROS observed in senescent cells may drive mtDNA damage creating a positive feedback loop leading to further increases in ROS and DNA damage highlighting the cyclical interactions between mitochondrial dysfunction, oxidative stress, and senescence, and illustrating how the initiation of any of these processes could lead to a downward spiral in tissue function. 

Perhaps unsurprisingly, given the high-volume density of mitochondria required to fulfil the heart’s energy demand, the heart has both high mtROS production and elevated mtROS, which have been shown to contribute to the pathophysiology of a variety of CVDs, including atherosclerosis, cardiac ischemia/reperfusion (IR) injury, HF, cardiac hypertrophy, and degenerative aortic valve disease [30,34,49,57]. ROS has also been shown to be a powerful inducer of senescence in multiple tissues and cell types, including the heart. Monoamine oxidase A (MAO-A) is a protein linked with driving oxidative stress; it is located at the outer mitochondrial membrane, involved in catalysing the oxidative deamination of monoamines, and produces hydrogen peroxide as one of its by-products [58]. Interestingly, cardiomyocyte-specific overexpression of MAO-A results in elevated ROS, increased senescence and mice display a dilated cardiomyopathy and myocardial dysfunction [59]. All of these can be rescued by treatment with antioxidants [4]. Similarly, the accelerated ageing mouse model *nfkb1^−/−^* showed increased ROS, telomere dysfunction, and cardiomyocyte hypertrophy [60]. Similar observations have been reported in more clinically relevant models: aged mice treated with the mitochondrial-targeted peptide SS-31 elamipretide had reduced myocardial ROS and improved cardiac function, which was associated with reduced senescence [61]. Furthermore, aged, senescent cardiomyocytes demonstrated an overall decline in the expression of most mitochondrial genes—particularly those genes involved in the ETC—and mitochondrial ultrastructural defects detected by transmission electron microscopy [61]. Myocardial infarction (MI) results in alterations to mitochondrial dynamics, culminating in increased ROS production and increased oxidative stress, particularly when followed by the clinical gold-standard treatment of reperfusion [62]. Several studies have shown that, even in this acute setting of increased oxidative stress, senescence is induced in multiple cell populations, including cardiomyocytes, and that these cells are active participants in post-MI myocardial remodelling since their elimination attenuates inflammation remodelling and improves functional outcomes [17,19]. 

Being so closely associated with cardiomyocyte dysfunction, mitochondrial damage is of major interest when exploring the mechanisms underpinning the cardiotoxicity of many otherwise beneficial therapeutics. This is relevant to both preclinical drug development, where cardiac liabilities remain a leading cause of drug attrition [63,64], but also to therapies approved for clinical use today which risk future withdrawal from the market due to cardiac adverse drug reactions (ADRs) [65]. Both traditional and new-generation oncology treatments are plagued by off-target cardiotoxic effects [66]. With cardiovascular disease being a leading noncancer cause of death in an ever-growing population of cancer survivors, understanding the mechanisms behind these cardiotoxicities is increasingly important [67]. As a case in point, anthracycline chemotherapies are notoriously chronically cardiotoxic and have been shown to deleteriously affect mitochondrial function in many ways.

Doxorubicin (DOX), an anthracycline commonly used in clinics, was historically shown to redox cycle via interactions with mitochondrial complex I, generating excessive ROS as a result [68,69]. Studies have subsequently demonstrated that DOX has a high affinity for cardiolipin, a lipid housed in the inner mitochondrial membrane which is essential for effective energy metabolism and proper mitochondrial function [70]. Notably, DOX becomes concentrated in the mitochondria of isolated neonatal rat cardiomyocytes, supporting the notion that this organelle is particularly vulnerable to off-target anthracycline toxicity [71]. Zhang and colleagues showed that mitochondrial function and oxidative phosphorylation pathways were disturbed in cardiomyocytes isolated from DOX-dosed mice, and that this was dependent on the topoisomerase IIβ (TopIIβ) enzyme which is thought to be crucial in the cardiotoxicity of this drug [72]. It has since been shown that DOX intercalates into mtDNA, which aids its accumulation in cardiomyocyte mitochondria in the same model [73]. Furthermore, Ichikawa and colleagues showed that DOX causes iron accumulation in cardiomyocyte mitochondria, leading to downstream toxicity. DOX treatment is associated with the depletion and mutation of mtDNA, as identified in the hearts of cancer patients [74]. The interplay between DOX-induced mitochondrial damage and cardiomyocyte senescence within this cardiotoxicity is less well-understood, but it has been shown that the two phenomena go hand-in-hand using in vitro and in vivo studies, as evidenced by Mitry et. al., amongst others [75,76,77]. As a long-established therapy, the impact of DOX upon cardiomyocyte mitochondria has been well reviewed [78], but newer oncology therapies are far less well-understood. For example, though tyrosine kinase inhibitors (TKIs) provide more targeted anticancer actions, the TKI sorafenib has historically been shown to impair cardiomyocyte mitochondrial function at clinically relevant doses in vitro, and more recent reports highlight that sunitinib may also induce cardiomyocyte mitochondrial damage via ROS accumulation [79,80]. Several other oncology therapies display off-target cardiovascular effects and mitochondrial toxicities [81] and it is clear the changing landscape of cancer survivorship necessitates thorough investigations into the long-term cardiac effects of both established and emerging antineoplastic therapies, and mitochondrial toxicity remains an attractive avenue of research.

### 2.2. Mitochondria Dynamics Imbalance

In physiological conditions, mitochondria constantly undergo co-ordinated cycles of fusion and fission, also referred to as mitochondrial dynamics [82]. Mitochondrial fusion is characterised by the union of two mitochondria resulting in one elongated mitochondrion, which allows for the dynamic repair of reversibly damaged mitochondria. Conversely, mitochondrial fission is characterised by the fragmentation of one irreversibly damaged and potentially harmful mitochondrion into small and spherical mitochondria that can be isolated and removed by mitophagy [83,84]. The coordination of these events is essential for the maintenance of mitochondrial quantity and quality, and therefore, the balance between them plays a vital role in the normal function of the cardiovascular system. Indeed, accumulating evidence has confirmed the influence of mitochondrial dynamics on the pathogenesis of CVD [82]. 

Mitochondrial fusion is first mediated by the transmembrane guanine triphosphatase (GTPase) proteins, mitofusin 1 (MFN1) and MFN2 in the outer mitochondrial membrane, and then by the optic atrophy protein 1 (OPA1) in the inner membrane [85]. Decreased levels of MFN1 and MFN2 have been found in animal models of atherosclerosis [86], and a decreased expression of OPA1 has been observed in post-MI hearts, which correlated with the downregulation of mtDNA and antioxidant genes [87]. Suggesting a causal role of fusion in CVD, ablation of the murine *Mfn1* and *Mfn2* genes in adult hearts induced mitochondrial fragmentation and dysfunction, and rapidly progressive and lethal dilated cardiomyopathy [88,89]. Different cardiac pathologies have also been associated with the formation of giant mitochondria or megamitochondria, as reviewed in [70], which evolve by fusion of the membranes of numerous large individual organelles due to the overexpression of protein fusion [90]. The opposing process, mitochondrial fission, is controlled by Mitochondrial fission protein 1 (Fis1) and Dynamin-related protein 1 (Drp1). It has been reported that Drp1 activation during cardiac IR results in left ventricular dysfunction and that Drp1 inhibition reduces cell death, preserves mitochondrial morphology, and inhibits the mitochondrial permeability transition pore [91,92]. While the relationship between mitochondrial fusion and myocardial senescence has yet to be investigated, elongated mitochondria have been observed in several senescent cell types, including fibroblasts, and are relevant to CVD, iPSC-derived and primary rat cardiomyocytes in vitro [93,94], and the depletion of *Fis1* mRNA levels leads to mitochondrial elongation, induces senescence, and increases ROS production [93,95]. 

Fusion and fission events control mitochondria biogenesis [96], a process that increases the number of mitochondria, improves the replication and repair of mtDNA, and induces the synthesis of mitochondrial enzymes and proteins [30]. The co-transcriptional regulator factor peroxisome-proliferator-activated receptor γ co-activator-1α (PGC-1α) induces mitochondrial biogenesis by activating the mitochondrial transcription factor A (TFAM), which drives the transcription and replication of mtDNA [97]. Reduced gene expression of PGC-1α has been associated with failing human hearts [98] and there is evidence that sirtuin-1 (SIRT1), a protein involved in metabolic regulation, delays the molecular characteristics of myocardial ageing by mediating the deacetylation of PGC-1α and the activation of mitochondrial biogenesis [99]. The *PGC-1α^+/−^/ApoE^−/−^* mouse model has shown that *PGC-1α* deficiency promotes vascular senescence, which is associated with increased oxidative stress, mitochondrial abnormalities, and reduced telomerase activity [100]. Mitochondrial biogenesis is also accompanied by variations in mitochondrial morphology [101]. Generally, various aspects of cardiovascular biology, including cardiac development, the response to cardiac IR injury and HF, are related to morphological and structural changes in mitochondria [87,91,102]. Dramatic changes in mitochondrial morphology have also been found in senescent cells, where mitochondria exist in a state of hyper-fusion as a response to reduced expression of mediators of the fission process and an overall reduction in the frequency of the fission and fusion events [103].

### 2.3. Cell Apoptosis and Mitophagy

Mitochondria are pivotal in controlling apoptosis, including the release of caspase activators and the participation of B-cell lymphoma-2 (BCL-2) family proteins [104,105]. Cardiomyocyte apoptosis plays a critical role in the pathogenesis and progression of all types of heart disease, particularly in ischemic heart disease and HF of various aetiologies [106]. For example, cardiac IR injury is related to the apoptotic death of cardiac muscle cells by activating the pro-apoptotic BCL-2 regulators BAX and BAK to change the integrity of the mitochondrial membrane and the cytosolic release of pro-apoptotic factors, which triggers caspase-dependent cell death [107]. In hypertension, the hormone angiotensin II, which plays an important role in volume and blood pressure control, has been linked to cardiomyocyte apoptosis in rats, and treatment with losartan has been associated with a reduction in cardiomyocyte apoptosis in both spontaneous hypertensive rats and hypertensive patients [108]. The subfamily of pro-apoptotic BCL-2 homology (BH) BH3-only proteins, BCL2/adenovirus E1B 19 kDa protein-interacting protein 3 (BNIP3) and its homologue BNIP3-like (BNIP3L or Nix), also induce apoptosis [109] and the forced expression of these genes is sufficient to induce cardiomyopathy in murine models [35,110,111]. Mitochondria are also important in other forms of cell death including necrosis. The mitochondrial permeability transition pore (MPTP), localized on the inner membrane of mitochondria, is the main player in oxidative stress-dependent cell death and increased mPTP is associated with ageing and age-associated disease [112]. In the context of the heart, mPTP activation is fundamental in causing myocardial damage following ischaemia-reperfusion (I/R) both as a result of myocardial infarction and transplantation. At the onset of ischaemia, oxidative phosphorylation is arrested due to a lack of oxygen which leads to the depolarisation of the mitochondrial membrane and loss of ATP. As the cellular metabolism rapidly shifts to anaerobic glycolysis, lactic acid is generated and the associated accumulation of hydrogen ions reduces intracellular pH levels, inhibiting myofibril contraction and closure of the mPTP. Upon reperfusion, the respiratory chain is rapidly exposed to oxygen, leading to oxidative stress, and Ca^2+^ accumulates due to rapid mitochondrial membrane potential restoration and pH is neutralized, which all contribute to the opening of the mPTP. The opening of the mPTP allows the free passage of molecules, including protons, through the inner mitochondrial membrane, uncoupling oxidative phosphorylation and disrupting ATP production. Impaired energy metabolism further results in a continuous cycle of increasing Ca^2+^ and mPTP causing osmotic swelling and damage and mitochondrial disruption and cellular necrosis [113]. Furthermore, this mitochondrial membrane disruption may also lead to a release of proapoptotic proteins, including cytochrome c, thereby also inducing apoptosis [112]. 

To prevent cardiomyocytes containing damaged mitochondria from undergoing apoptosis, mitophagy, a cargo-specific form of autophagy selectively targets the degradation of dysfunctional and damaged, and hence potentially cytotoxic, mitochondria within a cell [30]. There are two mechanisms described for mitophagy: adaptor-mediated and receptor-mediated. The former pathway functions via Phosphatase and Tensin Homolog (PTEN)-induced putative kinase 1 (PINK1) and Parkin-mediated mitophagy [114]. PINK1 is a serine/threonine kinase that continuously monitors mitochondrial health and provides a rapid response when mitochondrial function collapses [115]. When mitochondria lose membrane potential or amass unfolded protein, PINK1 accumulates on the outer membrane and both recruits and directly phosphorylates E3 ubiquitin ligase [115,116] or phosphorylates E3 ubiquitin via the intermediate phosphorylation of MFN2 [116]. The accumulation of ubiquitin in key mitochondria-associated proteins on the outer mitochondrial membrane, amplifies a signalling cascade involved in the recruitment of autophagosomes to target the damaged mitochondria. The mitochondria-containing autophagosome is trafficked to, and fused with, a lysosome and degraded [117]. In healthy young hearts, there is an underlying level of baseline mitophagy which is essential for maintaining cellular homeostasis in an energy-efficient heart, and for responding and adapting to stress [118]. However, decreased mitophagy is associated with CVD, as an accumulation of “old” defective mitochondria may reduce the heart’s potential to adapt to stress. Indeed, multiple animal studies have linked the deletion of mitophagy-related genes at the whole-body level or cardiomyocytes with the spontaneous development of cardiovascular disorders [119]. For example, mice bearing a cardiomyocyte-specific deletion of *Mnf2* prematurely succumbed to progressive cardiomyopathy, which could be partially reversed by restoring mitophagy in cardiomyocytes via the expression of the antioxidant enzyme catalase [120]. The whole-body *Pink1^−/−^* mice experienced left ventricular dysfunction and pathological cardiac hypertrophy by 2 months of age [121]. Mitophagy is also essential for reducing cardiac injury following MI. Under baseline conditions, *Parkin*-deficient mice hearts were shown to have smaller and disorganised mitochondria as revealed by ultrastructural analysis, but mitochondrial and cardiac function were unaffected [122]. However, after MI, these mice had reduced survival and developed larger infarcts when compared to control mice, which was associated with the rapid accumulation of dysfunctional mitochondria in the infarct border zone [122]. In patients with late-stage heart disease, a low number of autophagosomes in cardiomyocytes is associated with a poor prognosis [123]. Damaging events (e.g., acute cardiac IR injury) lead to the reduction in the autophagy flux, and as a consequence, damaged dysfunctional mitochondria accumulate in cardiomyocytes, leading to severe oxidative stress and apoptosis [124]. The destabilisation of atherosclerotic plaques has also been associated with deficient mitophagy [125,126]. Furthermore, a reduced expression of autophagic markers p62 and microtubule-associated protein light chain (LC3)-II has been detected within atherosclerotic plaques from human samples and mouse models [127,128]. In mouse models, the activation of mitophagy through antioxidant therapeutic strategies has been explored to stabilise atherosclerotic plaques [129]. 

Aside from the conventional forms of mitophagy, there are additional specialized pathways including a process which exhibits a notable level of specificity and involves mitochondrial-derived vesicles, as well as the selective removal of mitochondrial fragments containing specific cargo rather than the entire organelle. This mechanism relies on the coordination of mitochondrial dynamics, mitophagy, and the vacuolar protein sorting (VPS) or retromer complex. In this process, alterations to mitochondrial membrane potential and the oxidation state of mitochondrial sub-compartments induce membrane curvature. This, in turn, leads to the recruitment of PINK1 and Parkin. The retromer complex, comprised of the VPS26, VPS29, and VPS35 proteins, plays a crucial role by providing the force needed to generate a vesicle. Importantly, these vesicles are subsequently delivered to lysosomes or peroxisomes, and this delivery process operates independently of the autophagy proteins Autophagy related 5 or Microtubule-associated protein 1A/1B-light chain 3 [130]. It remains to be seen if changes in the dynamics of this non-canonical form of mitophagy are associated with CVD or senescence. 

Interestingly, despite and perhaps because of mitochondrial dysfunction, senescent cells express pro-survival pathways, enhancing survival and increasing resistance to apoptosis. Senescent cells are more resistant to apoptosis in response to stimuli, including serum withdrawal, ultraviolet damage, oxidative stress, and treatment with cytotoxic drugs [131]. While there is a heterogeneity between cell types and senescence stimuli, enhanced activation of several pathways including BCL-2 family members, p53/p21Cip, ephrins (EFNB1 or 3), the phosphatidylinositol-4,5-bisphosphate 3-kinase delta catalytic subunit (PI3KCD), plasminogen-activated inhibitor-1 and 2 (PAI1 and 2), and hypoxia-inducible factor-1α (HIF1α) can be involved [132,133,134] and are referred to as senescent cell anti-apoptotic pathways (SCAPs). As discussed below, the activation of these pathways may contribute to the proinflammatory nature of senescent cells.

## 3. Mitochondria Dysfunction, Senescence, and Inflammation in CVD

The role of inflammation in promoting CVD is increasingly recognised. Recent discoveries have demonstrated that mitochondria are key elements that stimulate innate immune signalling cascades which trigger inflammation and promote pathology in an expanding list of diseases, including cardiac pathologies [135]. Many investigations have revealed that, when mitochondrial integrity is compromised, mtROS and mtDNA act as damage-associated molecular patterns (DAMPs), endogenous molecules that are isolated within intracellular compartments and discharged to the extracellular space in response to damaged or dying cells [136], promoting pathological inflammatory responses by binding with pattern-recognition receptors (PRRs). For example, a study on mice found that mtDNA released by dying ischemic cells during MI activates the Interferon regulatory factor 3 (IRF3)-dependent innate immune response, which has a harmful effect on ventricular remodelling after MI [137]. The Stimulator of interferon genes (STING)-IRF3 pathway might also facilitate chronic inflammation and dysfunction in endothelial cells via sensing mtDNA [138], which are key events in the development of atherosclerosis and are associated with an elevated risk of many cardiovascular events [139]. Furthermore, cytoplasmic mtDNA which escapes from autophagy-mediated degradation cell-autonomously has been linked with the activation of the immune system via Toll-like receptor 9 (TLR9), which has been associated with elevated arterial pressure and vascular dysfunction in spontaneously hypertensive rats [140], and with exacerbated HF in mice [141]. Accumulating evidence has also shown that mtROS and mtDNA contribute to molecular inflammation events during the pathogenesis of CVDs, activating the Nod-like receptor (NLR) family, pyrin domain containing 3 (NLRP3) inflammasome [142,143], although how this unfolds remains unknown. For example, excessive mtROS and dysfunctional mitochondria are considered critical drivers responsible for NLRP3 activation during the progression of atherosclerosis, and the level of the inflammasome has been found to be highly associated with the severity of disease [144]. Upon activation, NLRP3 inflammasome activates caspase-1, which cleaves and matures the pro-inflammatory cytokines interleukin (IL)-1β and IL-18, which contribute to cardiac fibrosis and HF [145]. Elevated IL-1β levels have been also correlated with age-related CVD [143]. Furthermore, the suppression of NLRP3 extends the lifespan of obese adult mice by reducing liver steatosis and cardiac damage [146]. In turn, PRRs might also modulate mitochondrial dysfunction and apoptosis, protecting against mortality as occurs with the receptor NLR family member X1 (NLRX1) during IR injury [147].

A significant characteristic of senescent cells is the acquisition of a hypersecretory phenotype or SASP, a collection of many biologically active factors, such as inflammatory cytokines, chemokines, matrix remodelling proteases, extracellular vesicles, and growth factors [148]. This heterogeneous group of secreted proteins self-reinforce and spread senescence in an autocrine and paracrine manner, respectively, or affect the local tissue environment of senescent cells, and possibly, the entire organism [9]. Although some SASP factors are common to all senescent cells, its composition varies depending on the cell type and the nature of the stimulus [149]. In senescent cardiomyocytes, increased expression of SASP factors such as cellular communication network protein family member 1 (CCN1), interleukins (IL1α, IL1β, and IL6), tumour necrosis factor-alpha (TNFα), monocyte chemoattractant protein-1 (MCP1), endothelin 3 (Edn3), tumour growth factor-beta (TGFβ), and growth and differentiation factor 15 (GDF15) have been clinically linked with age-related myocardial ischemia and infarction [4,17,150]. mtROS are a component of the SASP [151], and functional mitochondria are critical for SASP production. As would be expected, senescent cells with depleted mitochondria have reduced ROS generation and also lose their proinflammatory phenotype yet remain in cell cycle arrest [77]. SASP production appears to involve mitochondria through several interconnected mechanisms. mtROS can induce c-Jun N-terminal kinase (JNK) signalling and the release of cytoplasmic chromatin fragments, triggering the innate immunity cytosolic DNA-sensing cyclic GMP-AMP synthase (cGAS)-STING pathway [152]. This, in turn, activates nuclear factor-κB (NFκB) signalling, switching on the transcription of proinflammatory genes and the SASP [152]. Recent studies suggest that expression of pro-survival pathways in senescent cells leads to sublethal apoptosis and minority mitochondrial outer membrane permeabilization (miMOMP). This miMOMP allows the release of mtDNA into the cytosol which activates the cGAS–STING pathway, resulting in the increased expression of inflammatory mediators and SASP [153]. The sublethal release of cytochrome c and caspase activation, associated with miMOMP may also contribute to further DNA damage, increased genetic instability, and perhaps deeper senescence [154]. Similarly, oxidative stress-dependent, but sub-lethal activation of mPTP may allow the release of solutes and ions, including superoxide, hydrogen peroxide, and calcium, resulting in damage to both mitochondrial and cellular proteins, lipids, and DNA, and thus accelerating cell aging [155].

## 4. Mitochondrial and Senescent Cells Targeted Therapies for CVD

### 4.1. Therapeutically Targeting Mitochondrial Dysfunction

In recent years, an increasing number of cardiac mitochondrial targets have shown their cardioprotective effects in experimental and clinical studies. Mitochondria are not only the site of OXPHOS; their dysfunction is also commonly associated with ATP deficiency and excessive ROS generation. Therefore, restoring their ATP-producing capacity and counteracting the damaging effects of ROS to reduce oxidative stress and chronic inflammation have been suggested as primary therapeutic targets to improve mitochondrial dysfunction [31]. 

AMP-activated protein kinase (AMPK) is an exclusive kinase of eukaryotes that plays a major role in regulating energy balance by monitoring changes in the level of intracellular ATP and coupling these changes to phosphorylation of downstream substrates, leading to an increase in ATP synthesis and/or a restriction of ATP depletion [156]. Thus, targeting the AMPK pathway has attracted widespread interest [30]. A well-known AMPK agonist is the first-line drug for treating Type 2 diabetes mellitus (T2DM), metformin [157]. Mechanistically, it has been widely accepted that metformin exerts beneficial effects through the inhibition of the respiratory chain at the level of Complex 1, leading to an increased AMP/ATP ratio and activation of the signalling kinase AMPK [158], which in turn, induces muscles to take up glucose from the blood. However, this mechanism requires higher doses of metformin than those used in the clinical routine [159]. Recently, it has been reported that clinically relevant doses of metformin activate AMPK through the lysosomal pathway, without perturbing AMP/ATP levels [159]. The authors demonstrated that metformin targeted lysosomal presenilin enhancer protein 2 (PEN2), which then inhibited the vacuolar H^+^-ATPase (v-ATPase) on the lysosome by binding to its AT6AP1 subunit. The formation of the PEN2-AT6AP1 axis initiates the lysosomal glucose-sensing pathway for AMPK activation [160]. Metformin also improves mitochondrial function and quality, as AMPK activation phosphorylates a range of target proteins involved in the regulation of mitochondrial biogenesis (PCG-1α), mitochondrial dynamics (Drp1, MFF), and mitophagy (PINK1-Parkin pathway) [161,162,163,164]. Interestingly, metformin has been described as senomorphic, being able to modulate SASP secretion from senescent cells and improve senescent cell function [165,166]. Metformin is inhibitory to SASP expression due to its inhibition of IκB kinase and IKKα/β phosphorylation, thereby preventing the NF-κB nuclear translocation [167]. Furthermore, metformin can attenuate senescence in human diploid fibroblasts and mesenchymal stem cells [168], and in healthy mice, metformin has been observed to extend health span and lifespan [169]. 

Indicating that metformin may have similar effects clinically, a recent meta-analysis has identified that patients with diabetes taking metformin have a significantly increased survival rate and a reduced incidence of age-related diseases, including CVD [170]. Several clinical studies have shown the beneficial effects of metformin in diabetes-related atherosclerosis, IR injury, and arrhythmia (as discussed by Bu et al.) [171]. For example, in the Reversing with MetfOrmin Vascular Adverse Lesions (REMOVAL) trial, a double-blind placebo-controlled randomized controlled trial to evaluate the cardiovascular effects of metformin in adults with T1DM, atherosclerosis progression was significantly reduced in metformin-treated patients [172]. In another randomized, placebo-controlled trial involving 390 patients with T2DM treated with insulin, metformin treatment improved endothelial function [173]. In non-diabetes patients, clinical studies are more controversial when evaluating the benefits of metformin in CVD. In a small clinical study consisting of 33 women who did not have diabetes, metformin reduced myocardial ischemia and improved endothelium-dependent microvascular responses in patients with angina, compared to the placebo [174]. However, in a subsequent clinical study consisting of 173 patients without diabetes with coronary heart disease treated with statins, Preiss et al. [175] found that metformin had no effect on disease progression and little or no effect on several surrogate markers of CVD. Although it remains unclear if the senomorphic activity of metformin contributes to any of these cardioprotective effects, in patients with carotid artery atherosclerosis, metformin ameliorates the proinflammatory state, which includes a reduction in serval SASP related proteins including IL-6 and TNF-α [176]. Metformin also influences other hallmarks of biological ageing. For example, nutrient-signalling pathways both through AMPK and SIRT1 activation, as well as downregulating insulin/insulin-like growth factor 1 (IGF-1) signalling and mechanistic targeting of rapamycin complex 1 (mTORC1). Metformin also attenuates oxidative damage and genome instability by enhancing DNA damage response and repair mechanisms, improving proteostasis by enhancing autophagy and inhibiting protein synthesis, and ameliorating mitochondrial dysfunction via mitochondrial complex I inhibition and PGC-1α upregulation. Furthermore, metformin treatment reduces telomere shortening by activating telomeric repeat-containing RNA [177]. Metformin is now being evaluated for its age-targeting effects in the TAME (Targeting Ageing with Metformin) clinical trial [178]. 

Because of the role of ROS in CVD, a reduction in oxidative stress through the supplementation with antioxidants such as Coenzyme Q10 (CoQ10), vitamin E, vitamin C, and β-carotene has been clinically studied in humans for the treatment of CVD, including HF, atherosclerosis, and acute MI [179,180,181,182]. However, with the sole exception of CoQ10, no clinically significant benefits were reported. CoQ10 is a lipid-soluble and biologically active quinone whose principal role is to participate in the ETC, where it functions as an electron carrier [183]. In a study including 420 patients with moderate to severe HF, Svend et al. [179] reported that long-term treatment with CoQ10, in addition to standard therapy, was safe, and associated with an improvement in symptoms and a reduction in mortality from cardiovascular events. Zeb et al. [180] reported beneficial effects on inflammatory markers and reduced progression of coronary atherosclerosis in patients treated with a capsule containing aged garlic extract and CoQ10 daily for 1 year. In a meta-analysis of 14 studies with 2149 enrolled subjects Lei et al. [184] found that patients with HF who used coQ10 had lower mortality. 

One possible explanation for the ineffectiveness of common antioxidants to show beneficial effects is their inability to enter the mitochondria, the primary source of ROS [185]. There are several approaches to targeting molecules towards the mitochondria and one of the most versatile is to develop synthetically modified antioxidants with lipophilic cationic compounds, such as triphenylphosphonium (TPP+) [186]. TPP+ is a membrane-permeant cation that is accumulated within the mitochondria up to several-hundredfold-fold because of the negative potential (-140 to –180 mV) generated across the inner mitochondrial membrane by the proton pumping action of the ETC [187]. Mitoquinone (MitoQ) is CoQ10 conjugated to TPP^+^ and has been shown to display impressive benefits in the treatment of CVD. It has been reported that 100 µM MitoQ in drinking water rescues the cardiac function of pressure-overloaded HF in a mouse model by decreasing hydrogen peroxide formation, improving mitochondrial respiration and mitochondrial permeability transition pore opening [188]. In mice and rat studies, MitoQ protected against IR injury by blocking oxidative damage within the mitochondria [189,190]. Treatment with MitoQ also prevented the development of hypertension, improved endothelial function, and limited cardiac hypertrophy in eight-week-old male spontaneously hypertensive rats [191]. Furthermore, MitoQ controls the expression levels of cardiac hypertrophy-associated transcript (*Chast*) and myosin heavy chain-associated transcript (*Mhrt*), two long non-coding RNAs involved in cardiac remodelling. It also attenuates adverse cardiac remodelling, and prevents HF in mice by inhibiting the interplay between TGF-β1 and mitochondrial-associated redox signalling [192]. In humans, a randomized, placebo-controlled, double-blind, crossover study of 20 healthy adults (60–79 years) with endothelial dysfunction demonstrated that oral supplementation with 20 mg/day of MitoQ was well tolerated and significantly improved endothelial function and reduced arterial stiffness and plasma-oxidized low-density lipoprotein (LDL), a marker of oxidative stress, through a reduction in mtROS [193]. Currently, in the USA, there are two ongoing clinical trials focused on the effects of MitoQ on cardiac function: the MitoQ Supplementation and Cardiovascular Function in Healthy Men and Women study (NCT03960073), and the Chronic Kidney Disease and Heart Failure With Preserved Ejection Fraction: The Role of Mitochondrial Dysfunction study (NCT03586414). Given that ROS is both an inducer and a consequence of senescence, it is perhaps unsurprising that, in a wide range of diseases and models, anti-oxidants (including those that are mitochondrial targeted) are demonstrated to attenuate senescence and SASP [194].

### 4.2. Senolytics, Senomorphics and Future Approaches

Despite active DNA damage responses, increased mitochondrial dysfunction, increased mitochondrial membrane permeability, and increased ROS production, senescent cells remain resistant to apoptosis. Based on these findings, Zhu and colleagues hypothesized that pharmacologically inhibiting the pro-survival networks could eliminate senescent cells [132]. This gave rise to the advent of compounds collectively termed senolytics, which target various components of anti-apoptotic pathways, including BCL-2 family members, to promote senescent cell apoptosis. 

In the context of heart health, studies have largely investigated the senolytic effects of the combination therapy dasatinib and quercetin (D&Q), and navitoclax (ABT-263). Dasatinib is a second-generation tyrosine kinase inhibitor, shown to inhibit ephrins, disrupting the pro-survival network that includes BCL-XL, PI3K, p21Cip, PAI1, and PAI2 [132,195,196]. Quercetin, a natural flavanol, inhibits multiple pro-survival proteins, including PAIs and PI3K, ultimately reducing BCL-W expression [197,198]. Navitoclax, a BH3 mimetic, induces senescent cell apoptosis by inhibiting anti-apoptotic proteins BCL-2, BCL-XL, and BCL-W [199,200].

Use of these senolytics in several animal models of CVD, such as age-related myocardial dysfunction, MI, anthracycline-induced cardiotoxicity and atherosclerosis, has provided proof-of-principle data that promoting mitochondrial mediated apoptosis in senescent cells reduces inflammation and attenuates disease pathophysiology [17,19,132,201,202,203,204]. On the other hand, as increased apoptosis has been implicated in age-related myocardial dysfunction, and after MI the primary objective of reperfusion therapy is to save as much myocardium as possible, concerns have been raised regarding the long-term outcomes of increased cell death [3]. However, recent studies have demonstrated that senescent cardiomyocytes are indeed detrimental to the outcome, and that the inhibition/modulation of the senescent phenotype may be beneficial in the senescent cardiomyocyte context: inhibition of *p16* in murine cardiomyocytes improved outcome following MI with reperfusion [3]. 

In the future, the modulation/inhibition of the senescent phenotype may be a more feasible route to successful intervention in this disease context, rather than the promotion of senescent cardiomyocyte apoptosis. As such, a senomorphic approach, modulating the senescent phenotype and attenuating the SASP, may consequently have more translational potential than senotherapies. Alternatively, specific inhibition of miMOMP-induced inflammation [153], may have therapeutic utility. For instance, the inhibition of the mitochondrial membrane BAX and BAK nanopores with small-molecule BAX inhibitor BAI1 was shown to decrease systemic inflammation and improve health span in aged mice [153]. Underpinning all these findings, the evaluation of a drug’s senolytic and/or senomorphic capabilities must be approached with rigour and caution, as emphasised in Niedernhofer and Robbins’ 2018 correspondence [204].

## 5. Conclusions

In the complex realm of cardiovascular diseases, mitochondrial dysfunction, and oxidative stress (in isolation or within the context of senescence induction and senescent cell function) play a key role in pathophysiology (Figure 3). A better understanding of these interconnected phenomena will enable the development of novel therapies for CVD. Ongoing research and clinical trials signal a new frontier in cardiovascular medicine, promising innovative treatments that could transform patient outcomes. These developments mark a significant stride towards enhancing both the quality and duration of life for those affected by cardiovascular diseases.

## Figures and Tables

**Figure 1 cells-13-00353-f001:**
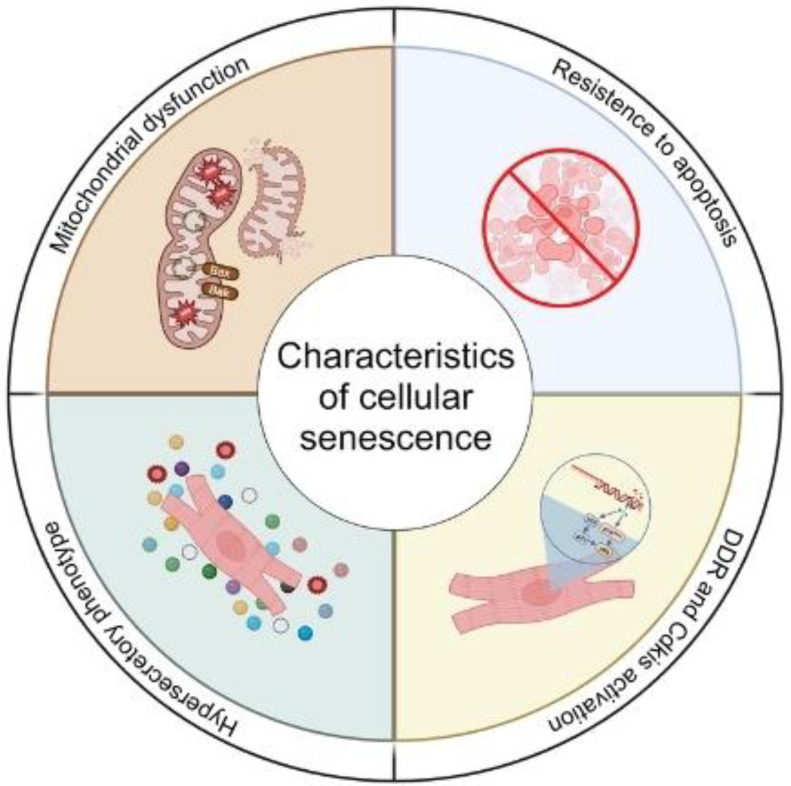
Common characteristics that define mitotic and post-mitotic senescent cells. DNA Damage Response (DDR) and Cyclin Dependant Kinase Inhibitors (Cdkis).

**Figure 2 cells-13-00353-f002:**
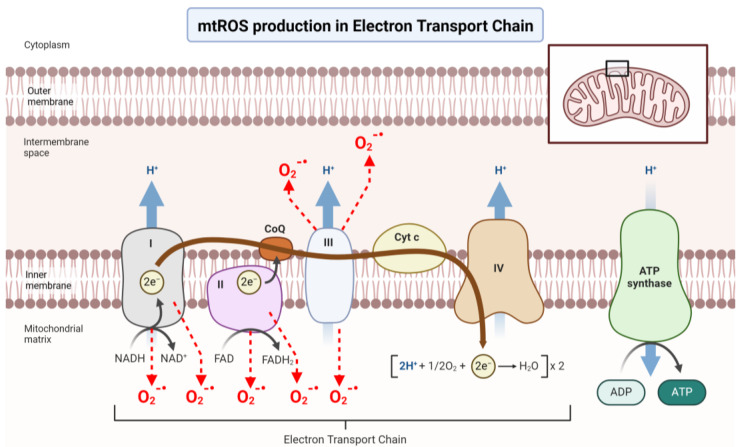
Illustrates mitochondrial reactive oxygen species (mtROS) production within the electron transport chain (ETC). Electrons, initially provided by NADH in complex I and FADH2 in complex II, traverse through ubiquinone to reach complex III. Subsequently, they move to complex IV via cytochrome c, where they combine with molecular oxygen to generate water. Proton-pumping activities by complex I, complex III, and complex IV into the intermembrane space establish a proton gradient crucial for ATP synthesis. During oxidative phosphorylation, electron leakage occurs, leading to the interaction with molecular oxygen and the formation of superoxide (O_2_**^−·^**). Complex I and complex III serve as the primary sites for ROS production within the mitochondria, while complex II also contributes. Complex III directs superoxide production both towards the matrix and the intermembrane space, whereas complex I and complex II exclusively produce ROS towards the matrix. Key components and molecules involved include coenzyme Q (CoQ), cytochrome c (Cyt c), electrons (e−), protons (H+), adenosine diphosphate (ADP), adenosine triphosphate (ATP), reduced (NADH) and oxidized (NAD+) nicotinamide adenine dinucleotide, flavin adenine dinucleotide (FAD), oxygen (O_2_).

**Figure 3 cells-13-00353-f003:**
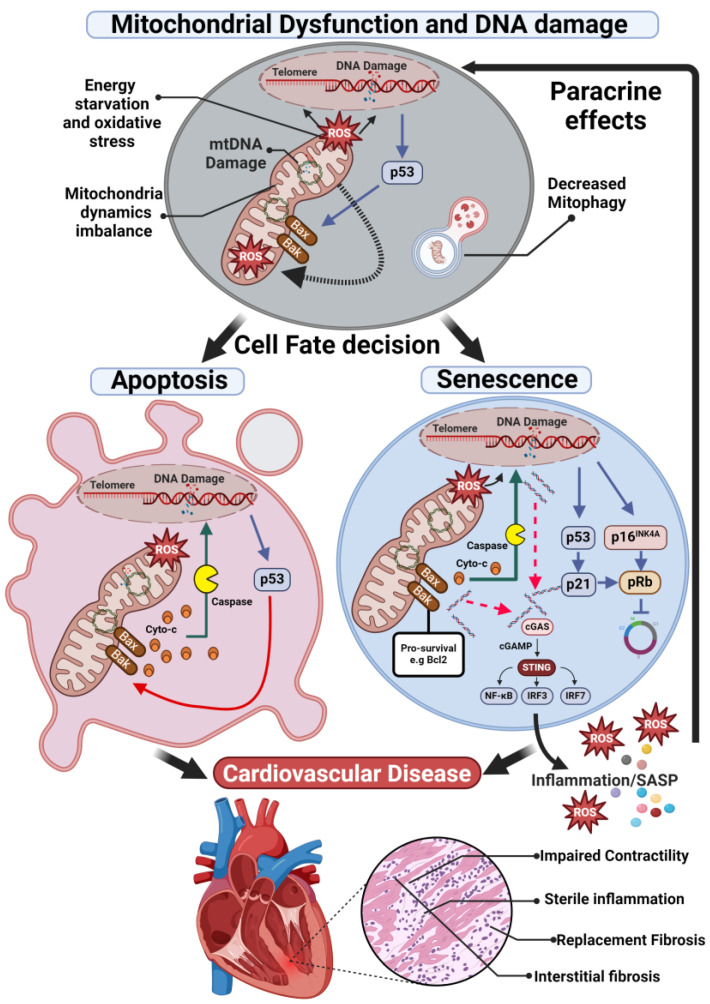
Mitochondrial dysfunction contributes to cardiovascular disease through apoptosis and senescence. Mitochondrial dysfunction and increased ROS production promote DNA damage, both chromosomal and mitochondrial. DNA damage leads to p53 activation and a cell-fate decision between apoptosis and senescence. In apoptotic cells p53 induces mitochondrial outer membrane permeabilization via formation of the apoptotic pore which allows cytochrome c release, activation of the caspase cascade, and cell death. While not yet completely understood, but perhaps as a result as less severe stress, DNA damage and p53 can lead to expression of the p21 (a negative regulator of apoptosis and the cell cycle), activation of the p16 pathway, or activation of both p21 and p16 pathways resulting in cellular senescence. Upregulation of pro-survival pathways in senescent cells suppresses apoptotic pore formation leading to miMOMP, sublethal apoptosis and the release of mtDNA into the cytoplasm. mtDNA fragments are sensed by the cGAS-STING pathway, upregulating expression of inflammatory mediators. Sublethal activation of the caspase cascade may also promote additional DNA damage. A combination of apoptosis and senescence will drive pathological myocardial remodelling.

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
