# Peer review of "Cellular Senescence, Mitochondrial Dysfunction, and Their Link to Cardiovascular Disease"

_cells, 2024, doi:10.3390/cells13040353_

Round 1
Reviewer 1 Report
Comments and Suggestions for Authors
In the reviewed manuscript, the authors M. Camacho-Encina et al. focus on cellular senescence in cardiovascular (CV) system and its implication for the CV diseases + the main cause of human death worldwide. Indeed, the progressive loss in balance between the CV cells apoptosis and loss of terminally differentiated cells as left ventricular cardiac myocytes merits further investigation. Specifically, the focus of this review to mitochondrial dysfunction, resistance to apoptosis and hypersecretory phenotype of CV cells represents actual and innovative avenue of research. Here, the authors provide robust evidence for the current understanding of these particular processes and demonstrate the importance of this research in practical treatment outcomes in latest clinical trials and in experimental medicine. Importantly, as authors declare in the text provided, the complex effect of CV system senescence involving the apoptosis, mtDNA quality control, ROS production, DNA repair is complex and tightly balanced and to maximize the patient treatment efficiency several if not all of reviewed mechanisms have to be considered and combined.
From the formal point of view, the text itself is robust and well written, the number and quality of the figures are appropriate. The aims as well the summary of the review are stated clearly, the abstract provides substantial information involved in the main text.
Considering all the positive points stated above and general benefit for the scientific community, from my point of view, there are following important issues which I consider as major comments for this manuscript:
1. Authors should clearly define what is the novelty of this manuscript in regard of the recently published review of the authors group (Booth et al., 2023, https://pubmed.ncbi.nlm.nih.gov/37120464/) as well as recent reviews of the groups of Chen et al., 2022 (https://pubmed.ncbi.nlm.nih.gov/33963378/) and Evangelou et al., 2023 (https://pubmed.ncbi.nlm.nih.gov/36049114/) where this manuscript, at least from my point of view, focus on very similar topic.
2. Several paragraphs of the review suffer from citations of the reviews and not original articles – by detailed analysis of the references, there is from 194 citations 101 cited reviews vs. 93 original papers. Especially the paragraph about Mitochondrial ATP production and ROS (lines 101-132, p. 3-4) is solely citing reviews and information focused on CV system (cells, experimental models, clinical evidence) is largely absent. Similar situation counts for chapter 1. Cellular senescence (lines 26-51, p. 1-2) and first paragraph of chapter 2. Mitochondrial abnormalities (lines 54-70, p. 2). Both paragraphs would greatly benefit from the reasoning provided by original articles and focused specifically on CV system.
The manuscript of authors M. Camacho-Encina et al. would further benefit from some minor recommendations:
1. The sentence at lines 228-231 (p. 5) would improve by providing exact information about the cells (ref. 85-87) from which this information has been extracted, namely if these are the CV or some other type of cells.
2. I recommend to implement the explanation of the abbreviations ‘DDR’ and ‘Cdkis’ in Fig. 1.
3. Increase the font size of the letters ‘ROS’ in Fig. 2.
4. In Fig. 2, I recommend to depict the cell nuclear membrane, as in the present form it looks like the cytochrome C and ROS are affecting the nuclear DNA directly.
5. MFN2 is beyond mitochondria located in the membrane of endoplasmic reticulum, I think for the general reader it would be useful to provide this information in the sentence at line 214 (p. 5).
After successful implementing of the comments mentioned above the manuscript of the authors M. Camacho-Encina et al. may be considered for publication in the Cells Journal.
Author Response
We would like to the thank the editor and reviewers for there time and consideration of our review. We have made every effort to address all concerns raised. Please see a point-by-point rebuttal below.
- Authors should clearly define what is the novelty of this manuscript in regard of the recently published review of the authors group (Booth et al., 2023, https://pubmed.ncbi.nlm.nih.gov/37120464/) as well as recent reviews of the groups of Chen et al., 2022 (https://pubmed.ncbi.nlm.nih.gov/33963378/) and Evangelou et al., 2023 (https://pubmed.ncbi.nlm.nih.gov/36049114/) where this manuscript, at least from my point of view, focus on very similar topic.
Thank you for your comment. To clarify, in the current review which we were invited to submit for the "Inflammation in Redox Modulation during Pathophysiology", as per our brief, we have focused on the role of the mitochondria in cardiovascular disease, including the interactions between mitochondrial dysfunction and senescence. As such, this review provides a novel oversight of aspects of mitochondrial function and dysfunction including Energy starvation and oxidative stress, Mitochondria dynamics imbalance Cell apoptosis and mitophagy as well as an overview of Therapeutically targeting mitochondrial dysfunction. These aspects of mitochondrial dysfunction and the contribution to the pathophysiology of cardiovascular disease are not discussed in detail in the reviews highlighted, which are more senescence focused.
- Several paragraphs of the review suffer from citations of the reviews and not original articles – by detailed analysis of the references, there is from 194 citations 101 cited reviews vs. 93 original papers. Especially the paragraph about Mitochondrial ATP production and ROS (lines 101-132, p. 3-4) is solely citing reviews and information focused on CV system (cells, experimental models, clinical evidence) is largely absent. Similar situation counts for chapter 1. Cellular senescence (lines 26-51, p. 1-2) and first paragraph of chapter 2. Mitochondrial abnormalities (lines 54-70, p. 2). Both paragraphs would greatly benefit from the reasoning provided by original articles and focused specifically on CV system.
Response: Thank you for the suggestion, we have now replaced several citations with those of the original studies and have included additional citations of original studies when appropriate.
- The sentence at lines 228-231 (p. 5) would improve by providing exact information about the cells (ref. 85-87) from which this information has been extracted, namely if these are the CV or some other type of cells.
Response: The originally cited study examined fibroblasts, we have now added this information along with details for an alternative study (PMID: 36493857) which demonstrated that elongated mitochondria is also a feature of senescent cardiomyocytes.
- I recommend to implement the explanation of the abbreviations ‘DDR’ and ‘Cdkis’ in Fig. 1.
Response: These have been added
- Increase the font size of the letters ‘ROS’ in Fig. 2.
Response: We have increased the font size.
- In Fig. 2, I recommend depicting the cell nuclear membrane, as in the present form it looks like the cytochrome C and ROS are affecting the nuclear DNA directly.
Response: The nuclear membrane has been added to the figure, which is now Figure 3.
- MFN2 is beyond mitochondria located in the membrane of endoplasmic reticulum, I think for the general reader it would be useful to provide this information in the sentence at line 214 (p. 5).
Response: Thank you for the comment, indeed MFN2 is expressed on the membrane of the ER. However, in this context MFN2 is involved in ER-mitochondria juxtaposition rather than fusion. We have attempted to identify a way to include this information, however, in our opinion this confuses the text. As such we would rather leave the text as it is. If the editor agrees that this information is required, we will include a larger paragraph specifically related to ER-mitochondria juxtaposition.
Reviewer 2 Report
Comments and Suggestions for Authors
In this manuscript, the authors summarized the mechanisms of senescence in CVD and clinical perspectives with a specific focus on mitochondrial dysfunction. Cellular senescence is increasingly recognized as a major factor in aging and multiple pathological conditions thus this review is timely and has the potential to provide valuable information on this complex process. While this review is interesting, it can be improved in the following aspects.
1. The authors covered MOMP regulated by BCL2 family proteins but not mitochondrial permeability transition pore (mPTP) dependent cell death pathway. Discussion on the roles of mPTP in aging-associated disease should be included (eg., PMID: 33418876).
2. PINK1 dependent Parkin recruitment through Mfn2 phosphorylation was discussed. However, it has been widely accepted that PINK1 phosphorylates ubiquitin to recruit and activate Parkin to damaged mitochondria (PMIDs; 24751536, 27291334). This should be discussed.
3. Line 335; Please make sure if reference # 129 demonstrated the role of circulating mtDNA (not intracellular mtDNA/TLR9 signaling?).
4. Recent evidence demonstrates that metformin activates AMPK through the lysosomal pathway (PMIDs: 27732831, 35197629, 37473813). Please discuss these new findings.
5. A recent study discovered a mechanism by which mitochondrial nucleoids are eliminated through an endosomal-mitophagy-related pathway, ameliorating mitochondrial dysfunction (PMID: 36344526). This should be mentioned.
Minor comments
1. line 58; "adenosyl triphosphate" should be "adenosine triphosphate".
2. Lines 224-226. Drp1 is italicized. this could be typos.
Author Response
We would like to the thank the editor and reviewers for there time and consideration of our review. We have made every effort to address all concerns raised. Please see a point-by-point rebuttal below.
Reviewer 1
- The authors covered MOMP regulated by BCL2 family proteins but not mitochondrial permeability transition pore (mPTP) dependent cell death pathway. Discussion on the roles of mPTP in aging-associated disease should be included (eg., PMID: 33418876).
Response: Thank you for highlighting these important studies. We have now included additional discussion page 7 line 305 to 325.
“Mitochondria are also important in other forms of cell death including necrosis. The mitochondrial permeability transition pore (MPTP), localized on the inner membrane of mitochondria, is the main player in oxidative stress-dependent cell death and increased mPTP is associated with ageing and age associated disease[112]. In the context of the heart, mPTP activation is fundamental in causing myocardial damage following ischaemia-reperfusion (I/R) both as a result of myocardial infarction and transplantation. At the onset of ischaemia, oxidative phosphorylation is arrested due to lack of oxygen which leads to depolarisation of the mitochondrial membrane and loss of ATP. As the cellular metabolism rapidly shifts to anaerobic glycolysis, lactic acid is generated and the associated accumulation of hydrogen ions reduces intracellular pH levels, inhibiting myofibril contraction and closure of the mPTP. Upon reperfusion the respiratory chain is rapidly exposed to oxygen, leading to oxidative stress and Ca2+ accumulates due to rapid mitochondrial membrane potential restoration and pH is neutralized which all contribute to opening of the mPTP. Opening of the mPTP allows the free passage of molecules, including protons, through the inner mitochondrial membrane, uncoupling oxidative phosphorylation and disrupting ATP production. Impaired energy metabolism further results in a continuous cycle of increasing Ca2+ and mPTP causing osmotic swelling and damage and mitochondrial disruption and cellular necrosis[113]. Furthermore, this mitochondrial membrane disruption may also lead to a release of proapoptotic proteins, including cytochrome c thereby also inducing apoptosis[112].
And
Page 10 Line 462 to 465
“Similarly, oxidative stress-dependent, but sub -lethal activation of mPTP may allow the release of solutes and ions, including superoxide, hydrogen peroxide and calcium resulting in damage to both mitochondrial and cellular proteins, lipids and DNA and thus accelerate cell aging (PMID: 28758328)”
- PINK1 dependent Parkin recruitment through Mfn2 phosphorylation was discussed. However, it has been widely accepted that PINK1 phosphorylates ubiquitin to recruit and activate Parkin to damaged mitochondria (PMIDs; 24751536, 27291334). This should be discussed.
Response: We have added the discussion below to page 7 lines 334 to 340.
“When mitochondria lose membrane potential or amass unfolded protein, PINK1 accumulates on the outer membrane and both recruits and directly phosphorylates E3 ubiquitin ligase[115,116] or phosphorylates E3 ubiquitin via the intermediate phosphorylation of MFN2[116]. Accumulation of ubiquitin onto key mitochondria-associated proteins on the outer mitochondrial membrane, amplifies a signaling cascade involved in the recruitment of autophagosomes to target the damaged mitochondria.”
- Line 335; Please make sure if reference # 129 demonstrated the role of circulating mtDNA (not intracellular mtDNA/TLR9 signaling?).
Response: We thank the reviewer for highlighting this inaccuracy and have now corrected the sentence (now lines 416 to 418) to read “Further, cytoplasmic mtDNA which escapes from autophagy-mediated degradation cell-autonomously has been linked with the activation of the immune system via Toll-like receptor 9”
- Recent evidence demonstrates that metformin activates AMPK through the lysosomal pathway (PMIDs: 27732831, 35197629, 37473813). Please discuss these new findings.
Response: We thank the reviewer for highlighting these important findings and a discussion about this has been added to page 10 line 481 to 491.
“Mechanistically, it has been widely accepted that metformin exerts beneficial effects through the inhibition of the respiratory chain at the level of Complex 1, leading to an increased AMP/ATP ratio and activation of the signaling kinase AMPK[158], which in turn, induces muscles to take up glucose from the blood. However, this mechanism requires higher doses of metformin than that used in the clinical routine[159]. Recently, it has been reported that clinically relevant doses of metformin activate AMPK through the lysosomal pathway, without perturbing AMP/ATP levels[159]. The authors demonstrated that metformin targeted lysosomal presenilin enhancer protein 2 (PEN2), which then inhibited the vacuolar H+-ATPase (v-ATPase) on the lysosome by binding to its AT6AP1 subunit. Formation of the PEN2-AT6AP1 axis initiate the lysosomal glucose-sensing pathway for AMPK activation[160].
- A recent study discovered a mechanism by which mitochondrial nucleoids are eliminated through an endosomal-mitophagy-related pathway, ameliorating mitochondrial dysfunction (PMID: 36344526). This should be mentioned.
Response: We thank the reviewer for highlighting this study, we have added discussion regarding this information to page 8, Lines 368-381.
This reads “Aside from the conventional forms of mitophagy, there are additional specialized pathways including a process which exhibits a notable level of specificity and involves mitochondrial-derived vesicles, and selective removal of mitochondrial fragments containing specific cargo rather than the entire organelle. This mechanism relies on the coordination of mitochondrial dynamics, mitophagy, and the vacuolar protein sorting (VPS) or retromer complex. In this process, alterations in mitochondrial membrane potential and the oxidation state of mitochondrial sub-compartments induce membrane curvature. This, in turn, leads to the recruitment of PINK1 and Parkin. The retromer complex, comprised of VPS26, VPS29, and VPS35 proteins, plays a crucial role by providing the force needed to generate a vesicle. Importantly, these vesicles are subsequently delivered to lysosomes or peroxisomes, and this delivery process operates independently of the autophagy proteins Autophagy related 5 or Microtubule-associated protein 1A/1B-light chain 3 [130].It remains to be seen if changes in the dynamics of this non-canonical form of mitophagy are associated with CVD or senescence.”
Minor comments
- line 58; "adenosyl triphosphate" should be "adenosine triphosphate".
Response: Corrected.
- Lines 224-226. Drp1 is italicized. this could be typos.
Response: Corrected.
Reviewer 3 Report
Comments and Suggestions for Authors
In this nice review, authors focus on cellular senescence and mitochondrial dysfunction, which contribute to CVD. Mitochondrial ROS generation is one of main reasons of which affect cellular functions.
Questions:
1. In the rewiev (2.1 paragraph) authors explained ATP synthesis in mitochondria in physiological and HF conditions. Therefore, I missed the information how mitochondria are involved in ROS synthesis and how alterations in mitochondrial functions in HF are related to ROS synthesis?
2. Mitochondrial ROS targets and damage of mitochondria and cell after exposion to increased ROS in HF?
Author Response
We would like to the thank the editor and reviewers for there time and consideration of our review. We have made every effort to address all concerns raised. Please see a point-by-point rebuttal below.
Questions:
- In the rewiev (2.1 paragraph) authors explained ATP synthesis in mitochondria in physiological and HF conditions. Therefore, I missed the information how mitochondria are involved in ROS synthesis and how alterations in mitochondrial functions in HF are related to ROS synthesis?
Response: We thank the reviewer for the comment and agree this information should be clearer, we have therefore added a new Figure to page 4 (Figure 2), which illustrates mitochondrial reactive oxygen species (mtROS) production within the electron transport chain (ETC).
- Mitochondrial ROS targets and damage of mitochondria and cell after exposion to increased ROS in HF?
We are very sorry but are unsure what the reviewer is requesting in this comment.
Round 2
Reviewer 1 Report
Comments and Suggestions for Authors
The authors M. Camacho-Encina et al. made significant progress to improve the quality of the manuscript submitted. Beyond the comment below, all the raised points were answered adequately and the formal issues were corrected. Therefore I consider the status of the review process as minor changes.
- Authors should clearly define what is the novelty of this manuscript in regard of the recently published review of the authors group (Booth et al., 2023, https://pubmed.ncbi.nlm.nih.gov/37120464/) as well as recent reviews of the groups of Chen et al., 2022 (https://pubmed.ncbi.nlm.nih.gov/33963378/) and Evangelou et al., 2023 (https://pubmed.ncbi.nlm.nih.gov/36049114/) where this manuscript, at least from my point of view, focus on very similar topic.
Thank you for your comment. To clarify, in the current review which we were invited to submit for the "Inflammation in Redox Modulation during Pathophysiology", as per our brief, we have focused on the role of the mitochondria in cardiovascular disease, including the interactions between mitochondrial dysfunction and senescence. As such, this review provides a novel oversight of aspects of mitochondrial function and dysfunction including Energy starvation and oxidative stress, Mitochondria dynamics imbalance Cell apoptosis and mitophagy as well as an overview of Therapeutically targeting mitochondrial dysfunction. These aspects of mitochondrial dysfunction and the contribution to the pathophysiology of cardiovascular disease are not discussed in detail in the reviews highlighted, which are more senescence focused.
I am thankful the authors for their answer, now the point for me is clear. As the authors provide this detailed answer, I would recommend to implement this arguing of the originality of this review to the abstract to be clear for the general audience as well, for example to modify the sentence:
In this review, we focus on cellular senescence and mitochondrial dysfunction, which have long been established to contribute to CVD. We also assess the recent advances in targeting mitochondrial dysfunction and senescence with a focus on therapies that influence both and therefore perhaps represent strategies with the most clinical potential, range, and utility.
to
In this review, we focus on cellular senescence and mitochondrial dysfunction, which have long been established to contribute to CVD. We also assess the recent advances in targeting mitochondrial dysfunction including energy starvation and oxidative stress, mitochondria dynamics imbalance, cell apoptosis, mitophagy and senescence with a focus on therapies that influence both and therefore perhaps represent strategies with the most clinical potential, range, and utility.
Author Response
Thank you for this suggestion, we have made your suggested changes to the abstract and agree that this better defines the novelty of the review.
